# Understanding the fine-scale heterogeneity and spatial drivers of malaria transmission in Kenya using model-based geostatistical methods

**Donnie Mategula** [1,2,3☯] *, **Judy Gichuki** [4,5☯]

**1** Malawi-Liverpool-Wellcome Programme, Blantyre, Malawi, **2** Liverpool School of Tropical Medicine, Liverpool, United Kingdom, **3** Kamuzu University of Health Sciences, Blantyre, Malawi, **4** Strathmore University, Institute of Healthcare Management, Nairobi, Kenya, **5** Health Services Department, Nairobi City County Government, Nairobi, Kenya

☯ These authors contributed equally to this work.
* dmategula@mlw.mw

**Data Availability Statement:** The datasets used and/or analyzed during the current study are available upon request from the Demographic Health Survey Programme Team. The request can

## Abstract

Malaria remains a public health concern. Monitoring the fine-scale heterogeneity of the malaria burden enables more targeted control efforts. Although malaria indicator surveys (MIS) have been crucial in evaluating the progress of malaria control interventions, they are only designed to provide a cross-sectional national and regional malaria disease burden. Recent advances in geostatistical methods allow us to interpolate national survey data to describe subnational disease burden that is crucial in informing targeted control. A binomial geostatistical model employing Markov chain Monte Carlo (MCMC) parameter estimation methods is used to understand the spatial drivers of malaria risk in Kenya and to predict malaria risk at a fine-scale resolution, including identifying hotspots. A total of 11,549 children aged six months to 14 years from 207 clusters were sampled in this survey and used in the present analysis. The national malaria prevalence based on the data was 8.4%, with the highest in the lake endemic zone (18.1%) and the lowest in the low-risk zone (<1%). The analysis shows that elevation, proportion of insectcide treated net (ITN) distributed, rainfall, temperature and urbanization covariates are all significant predictors of malaria transmission. The 5x5 Km resolution maps show that malaria is heterogeneous in Kenya, with hotspot areas in the lake endemic area, the coastal areas, and some parts of the shores of Lake Turkana and Kajiado. The high-resolution malaria prevalence maps produced as part of the analysis have shown that Kenya has additional malaria hotspots, especially in areas least expected. These findings call for a rethinking of malaria burden classification in some regions for effective planning, implementation, resource mobilization, monitoring, and evaluation of malaria interventions in the country.

be made via the link https://dhsprogram.com/Data/
.

**Funding:** The authors received no specific funding
for this work.

**Competing interests:** The authors have declared
that no competing interests exist.

## Background

Malaria remains a public health concern and continues to be one of the most important tropical diseases affecting human populations to date [1]. In 2020, an estimated 241 million malaria cases occurred worldwide, 90% of which were in sub-Saharan Africa [2]. The disease is caused by protozoa of the genus *Plasmodium* of which five known species, *Plasmodium falciparum*, *P. vivax*, *P. ovale*, and *P. malariae*, more recently, *P. knowlesi*, are responsible for human Infection [3,4]. The vector responsible for human transmission is the female anopheles mosquito. In Sub-Saharan Africa, malaria is one of the leading causes of morbidity and mortality, especially in children under five. Other high-risk groups include pregnant women and immunologically naïve persons like travellers coming from non-endemic places [5].

Malaria remains a significant public health problem in Kenya, accounting for an estimated 13% to 15% of outpatient cases, with nearly 70% of the population at risk for malaria [6]. Four out of the five species of malaria parasite that cause human Infection are present in Kenya, but the *Plasmodium falciparum* parasite is the predominant cause of Infection in the country. Over the past decade, Kenya has substantially scaled up available malaria control tools, such as insecticide-treated bed nets, indoor residual spraying and the use of artemisinin-based combination therapies [6]. Evidence of this massive scale-up of interventions is the observed decline in prevalence. Kenya has experienced a decrease in the national prevalence of malaria among children ages six months to 14 years, from 13 per cent in 2010 to 8 per cent in 2015, and % in 2020 [6].

One of the key objectives of the Kenya Health Policy 2014–2030 is the elimination of communicable diseases, including malaria. This is supported by the Kenya malaria strategy for 2019 to 2023, which sets a vision of a malaria-free Kenya and targets to reduce malaria incidence and mortality by seventy-five per cent by 2023, with 2016 as the baseline year[6].

Malaria transmission in Kenya varies geographically. This could be due to varied climatic conditions, vector and parasite resistance, differences in intervention uptake across populations and other unmeasured factors that are thought to be responsible for this increasing heterogeneity [6,7]. The country is administratively divided into five malaria epidemiological zones based on risk profiles. These zones include highland epidemic-prone areas, lake endemic areas, coast endemic, semi-arid seasonal, and low-risk malaria areas. The endemic areas lie in the lake and coastal regions with altitudes ranging from 0m to 1300 above sea level. These areas have perennial malaria transmission due to rainfall, temperature, humidity and other critical factors that drive malaria transmission. The semi-arid seasonal malaria transmission areas are in the country's northern, northeastern, and southeastern parts. These areas experience short periods of intense malaria. The highland epidemic-prone areas are located within the western highlands and have seasonal malaria transmission with some yearly variation. The altitude in these zones is relatively higher than the other zones, lying 1500 meters above sea level. The malaria epidemics in the highland epidemic-prone zones are less predictable. Lastly, the low-risk malaria areas cover Nairobi and the central highland. Temperatures are usually too low to allow the completion of the sporogony cycle of the malaria parasite in the vector in the low-risk zones [6].

Malaria indicator surveys (MIS) measure progress on key malaria indicators in Kenya. The country has conducted four MIS in 2007, 2010, 2015 and 2020. The MIS are nationally representative household surveys that provide estimates of national and regional malaria indicators to assist malaria control programs in tracking their progress and evaluating the impact of strategies and interventions. The MIS follow a standard methodology recommended by the Roll Back Malaria Monitoring and Evaluation working group guidelines [8]. Originally, MIS surveys were designed to measure the blanket scale up of interventions like bed nets, using a

classic two-stage sample design and coverage indicators as the primary endpoints. Over time, as coverage increased, interest expanded to the impact of parasite prevalence. While it is still a norm that the MIS traditionally measures progress in these areas, the survey methodologies need to consider the underlying disease heterogeneity [8]. Recent advances in statistical analyses, including geostatistics, have made it possible to make fine-scale inferences of malaria transmission from survey data like the MIS, that is not traditionally designed for such extrapolations. In this paper, we use the Kenya 2020 MIS data to achive the following objectives:

i. understand the relationship between malaria prevalence and several factors including environmental factors and

ii. understand the disease heterogeneity across the country's surface including identification of hotspots.

 We hypothesized that:

i. **T**here is a significant association between environmental factors (such as temperature, humidity, and rainfall patterns) and malaria prevalence in different regions of Kenya as evidenced from the 2020 Malaria Indicator Survey (MIS) data.

ii. There is substantial heterogeneity in malaria prevalence across different regions of Kenya, with identifiable hotspots of higher prevalence, as can be determined from an analysis of the 2020 MIS data.

iii. The existing methodology of MIS, primarily focused on intervention coverage indicators, might be inadequate in capturing the evolving complexities and heterogeneities of malaria transmission in Kenya, and propose that model based geostatics has the potential to capture this very well.

## Methods

### Country profile

Kenya is an East African country that covers an area of 582,550 km$^2$. It is bordered by Ethiopia to the north, Tanzania to the south, Uganda to the west, South Sudan to the northwest, and Somalia to the northeast. Approximately 80% of Kenya's land is arid and semi-arid, only 20% is arable, and only 1.9% of the total surface area is occupied by standing water. The great East African Rift Valley extends from Lake Victoria to Lake Turkana and further southeast to the Indian Ocean [9]. The country has a number of large rivers including the Tana, Galana, Turkwel and Nzoia [10]. Fig 1 below is a map of Kenya showing the five epidemiological zones as defined by the national malaria program [6].

### Data

This secondary analysis used data from the Kenya MIS [6]. Access to the dataset was given to the authors on Apr 28 2022. The datasets were de-identified. The Institutional Review Board (IRB-)approved procedures for Demographic Health Survey(DHS) public-use datasets do not in any way allow respondents, households, or sample communities to be identified. Authors had no access to the names of individuals or household addresses in the data files. Additionally, the geographic identifiers only go down to the regional level, which is hard to identify individuals.

 The 2020 MIS, the fourth conducted by the country, followed a similar design and set-up as the former ones. It was conducted during the peak malaria season in November and December

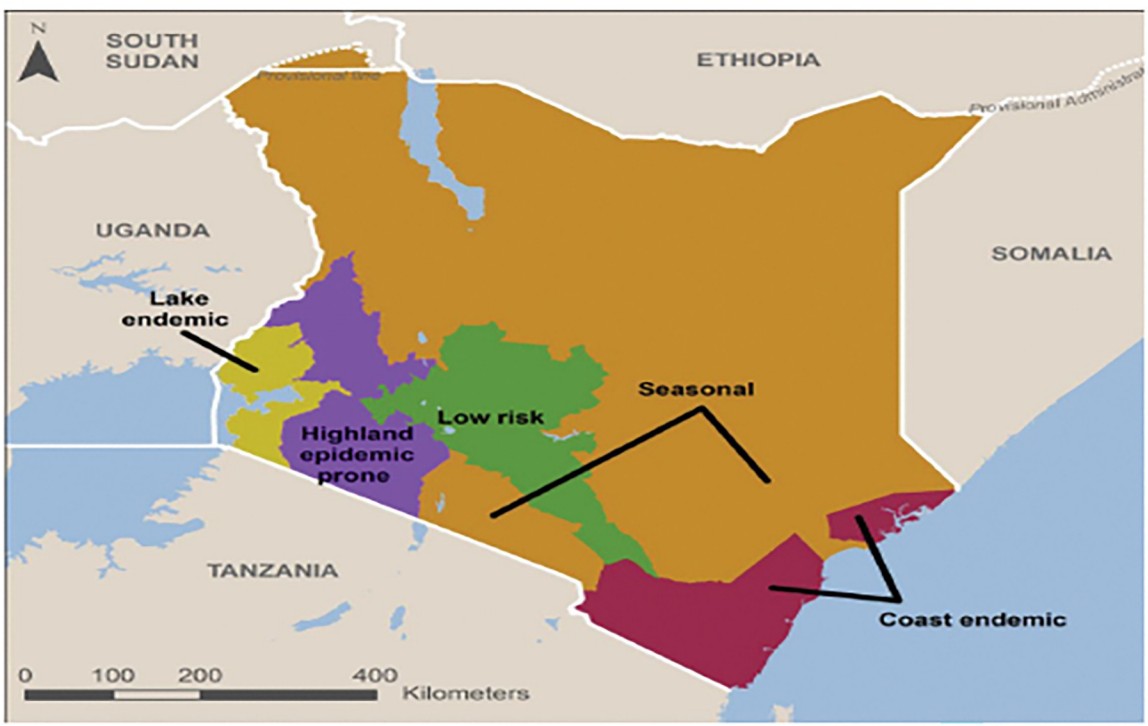

**Fig 1. Kenya epidemiological malaria zones.**

2020. A two-stage stratified sampling design was used, powered to give malaria parasite prevalence estimates and other key malaria indicators at the national level (urban and rural areas) and for the five epidemiological zones. The first stage sampling unit was a cluster developed from enumeration areas (EAs). EAs are the smallest geographical areas created for purposes of census enumeration. The EAs used were based on the 2019 Kenya population census. In the Kenya MIS, a cluster was defined as either an EA or part of an EA. A total of 301 clusters (134 urban and 167 rural) were sampled in this first stage using the probability proportional to size approach. The second stage sampling unit was households. In each cluster, 30 households were selected from a line listing of the sampled clusters using a systematic random sampling approach. A total of 7,952 households were sampled. All women aged 15–49 in the selected households were eligible for individual interviews. They were asked questions about preventing malaria during pregnancy and treating childhood fevers. In addition, the survey included testing for anaemia and malaria among children aged six months to 14 years using a finger- or heel-prick blood sample.

Permission to use the dataset was obtained from The Demographic and Health Surveys (DHS) Program through the archiving office. The original study received ethical clearance from the Kenyatta National Hospital/University of Nairobi Scientific and Ethics Review Committee. All participants provided oral informed consent.

## Variables

**Outcome variable.** In this analysis, the outcome variable was a binary outcome derived from the total number of children tested and the total number testing positive. This was extrapolated to estimate the cluster-level plasmodium falciparum malaria prevalence (*Pf*PR).

**Explanatory variables.** The explanatory variables included cluster-level factors such as rainfall, temperature, elevation, and urbanization and individual-level characteristics such as gender and age. These variables have been shown to affect malaria transmission [11].

## Data cleaning

The data cleaning and analysis were done in R [12]. Maps produced in the analysis were further processed for better visualization in QGIS (Version 3.2). Relevant variables were extracted and renamed to shorter names for ease of coding. Coordinates in the initial dataset were given the longitude and latitude system, which were transformed into the universal coordinate system (UTM). All distances were scaled to kilometres.

## Exploratory analysis

The initial exploratory analysis was descriptive to understand the data and to explore the initial relationships between the outcome variable of prevalence with the covariates in the data set. Scatter plots with fitted linear regression lines were used for this step to observe the relationship between prevalence and the explanatory variables. To further understand the variables, correlation plots were used to understand the relationships between the variables to guide the decisions of which covariates to include in the Model. The additional exploratory analysis involved plotting the clusters on the Kenyan map's surface, showing the sampled cluster's distribution and the crude malaria prevalence.

## Model fitting

The first objective of the analysis was to understand the relationships between malaria prevalence and several factors, including environmental factors. Several steps were followed:

1. Fitting a generalized linear model

2. Assessing evidence of residual correlation

3. Fitting a generalized linear mixed model

4. Reassessing evidence of residual correlations

5. Fitting a binominal geostatistical model and parameter estimation

6. Model validation

The model description for the generalized linear Model and the generalized linear mixed Model are described in S1 File.

## Description for the binominal geostatistical model

A detailed description for the model based geostatistics developed by Diggle and Giorgi are described elsewhere [13]

Let $Y_i$ denote the number of individuals that test positive for plasmodium falciparum at survey cluster location $x_i$

And that the survey team went to the sampled clusters given by $x_i$ and sampled $m_i$: $i = 1....$ $n$ individuals at risk in the cluster and recoded the outcome of every person that tests positive and negative for plasmodium falciparum malaria.

Then standard geostatistical Model assumes that:

$$Y_i \sim Binomial\ (m_i, P(x_i))$$

$Y_i$ is a Binomial distribution with $m_i$ trials and probability of a positive test $P(x_i)$ specified in the binomial geostatistical Model below:

$$log\left\{\frac{P(x)}{1 - P(x)}\right\} = \alpha + d(x_i)^T \beta + S(x) + Z_i$$

Where $\alpha$ is the intercept parameter, $S(x)$ are the spatial random effects, representing the spatial variation between the sampled clusters. $Z_i$ are mutually independent zero-mean Gaussian random variables with variance $r$ and in this analysis represent the spatial variation within cluster variation, measurement error or small-scale spatial variation.

$d(x_i)^T$ is a vector of observed spatially referenced explanatory variables associated with the response $Y_i$, and $\beta$ is a vector of spatial regression coefficients for the covariates.

The Matérn correlation function for the stationary Gaussian processes $S(x)$ used in this analysis, a two-parameter family is given by:

$$p(u, \varphi, k) = 2^{(k-1)}(u/\varphi)^k\ K_K + (u/\varphi)$$

Where:

- $u$ denotes the distance between two locations x and x′,

- $\varphi >0$ is a scale parameter that determines the rate at which correlation decays to 0 as the distance increases, and

- $k >0$, is a smoothness parameter which determines the analytic smoothness of the underlying process $S(x)$.

In the binomial geostatistical regression for this analysis, the Matérn shape parameter $k$ was set to 0.5 variance parameters $\tau^2$ to 0.

The covariates $d(x_i)^T$ used in the binomial geostatistical Model for prediction were obtained from an exploratory analysis set to understand the relationship of the variables with the outcome variable of malaria prevalence. This Model included the covariates: elevation, ITN usage, mean temperature, rainfall, and cluster urbanization (urban vs rural). The Markov chain Monte Carlo (MCMC) methods were used for parameter estimation in this Model. Confidence intervals of the estimates are calculated on the log scale then transformed back to the non-log scale that is used to report the results.

To test whether there was any evidence against spatial correlation in the data, empirical variogram methods are used. A simulation of 1000 empirical variograms around the fitted Model is ran and these are used to compute 95% confidence intervals at any given spatial distance of the variogram. A conclusion is reached that that there is a spatial correlation in the data if the empirical variogram obtained from the data falls outside the 95% tolerance bandwidth.

The second objective is to understand the disease heterogeneity across the surface of the country, including identification of hotspots and the uncertainty attached to these hot spots. For this purpose, a binomial geostastical model was used as described above but with covariates that were available as raster. These included urbanization, temperature, and precipitation. The target for the predictions was a prevalence of malaria over the 5 x 5 km regular grid surface covering the whole surface of Kenya. A map of malaria prevalence was generated. Uncertainty of the prevalence was addressed using *Exceedance Probabilities*, an approach that is more relevant to policy makers, than the traditional approach of using confidence intervals. *Exceedance*

**Table 1. Malaria prevalence in Kenya across five epidemiological zones.**

| Epidemiological Zone | Total clusters | Number tested(N) | Number positive(n) | Prevalence (n/N)% |
|---|---|---|---|---|
| Coastal Endemic | 29 | 1088 | 59 | 4.95 |
| Highland Epidemic Prone | 55 | 2122 | 33 | 1.56 |
| Lake Endemic | 97 | 4621 | 836 | 17.93 |
| Low Risk | 54 | 1307 | 1 | 0.08 |
| Seasonal | 56 | 2210 | 33 | 1.49 |

*Probabilities (EP)* method sets policy relevant thresholds. The *EP* can be formally expressed as:

$$EP = Probability \{, P(x_i) > t | \text{data}\}$$

where *t* is the prevalence threshold, set to 10% in this analysis.

## Results

A total of 11,549 children aged six months to 14 years were sampled. The analysis used 297 clusters. The number of clusters per transmission zone is shown in Table 1 below. The lake endemic area had the greatest number of clusters (97), while the coastal endemic area had the lowest number of clusters (29). The national malaria prevalence based on the data was 8.4%, with the highest in the lake endemic zone (18.1%) and the lowest in the low-risk zone (<1%).

The map in Fig 2 below shows the sampled locations and the cluster-level malaria prevalence on. The lake endemic area has the highest number of clusters sampled and is also the zone with the highest prevalence estimates at the cluster level.

The weather pattern varies across the surface of Kenya. The maps in Fig 3 below show the variation in temperature across space on the top, and the variation in annual precipitation for the year 2020 on the bottom.

### Binomial geostatistical model results

The binomial geostatistical Model results indicate that the elevation, proportion of ITN distributed, rainfall, temperature and urbanization covariates are all significant predictors of malaria transmission (Table 2).

### Model validation

The odds of malaria transmission are less in the urban clusters compared to the rural ones. Urban clusters have nearly 68% less malaria prevalence than rural ones (OR 0.32 CI: 0.26–0.39, P value <0.0001). The higher the rainfall, the higher the risk of malaria transmission. Every mm increase in the average rain increases malaria prevalence by 1.9 times (OR 1.91 CI 1.69–2.15, P value <0.0001). Rise in mean temperature also increases the risk of malaria prevalence. Every degree increase in temperature increases the odds of malaria prevalence by 1.4 times (OR 1.37CI 1.28–1.47, P value <0.0001).

Using variogram-based techniques described above, the Model above was tested for evidence of spatial correlation. The results of this process are shown in Fig 4 below. Since the empirical semi-variogram (solid line) falls within the 95% confidence interval (grey envelope), this shows that the Model is valid; the Model for malaria prevalence is compatible with the data.

Map of the sampled locations and their malaria prevalence

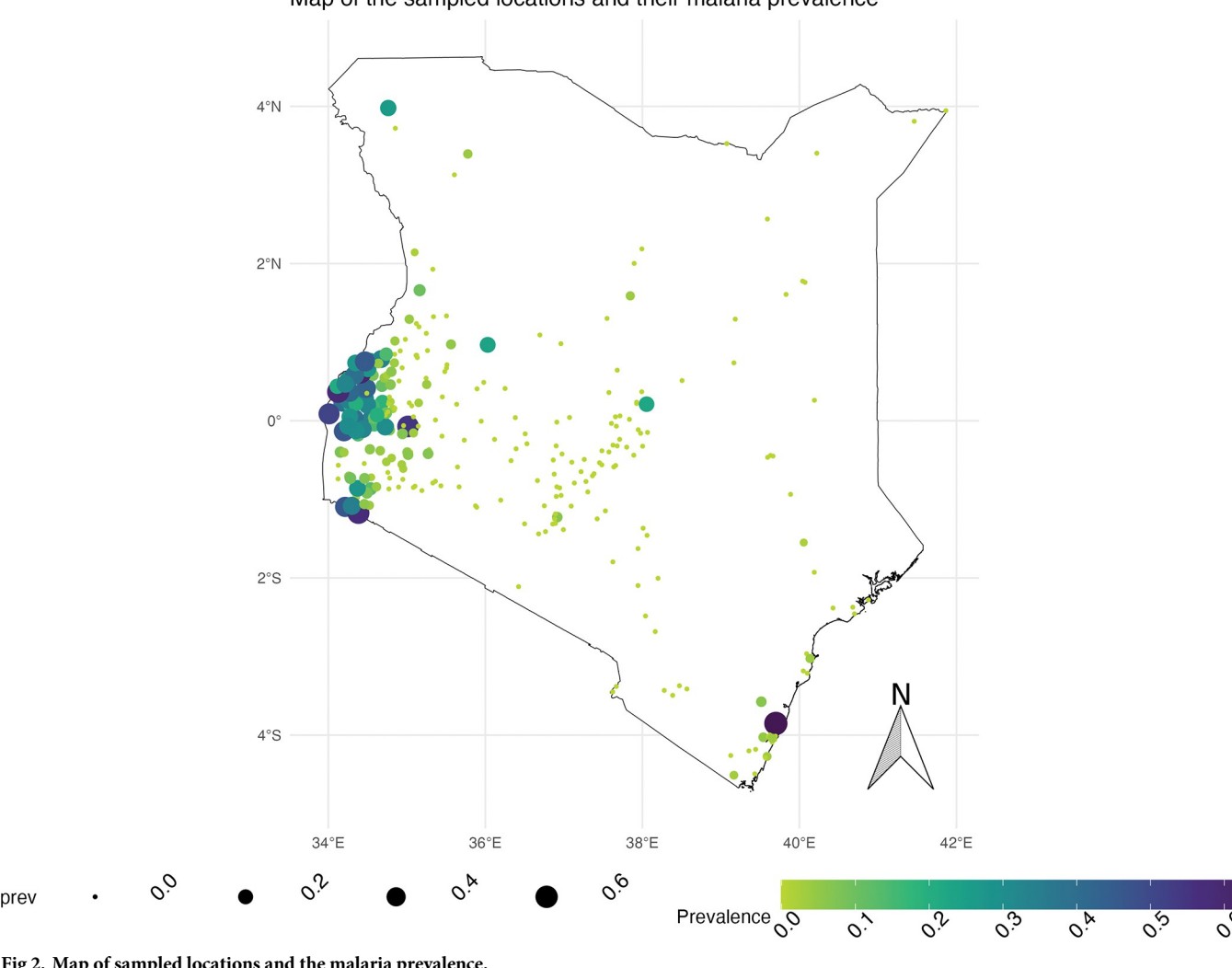

**Fig 2. Map of sampled locations and the malaria prevalence.**

## Prediction

To understand the disease heterogeneity across the country's surface, including the identification of hotspots, a 5 × 5 km resolution map for malaria prevalence in children six months to 14 years is presented in Fig 5 below. Overall, malaria prevalence is low in most parts of the country. Hotspots were notable in Western Kenya in the lake endemic areas around Lake Victoria, in the endemic coastal regions along the Indian Ocean and three hotspot areas within the seasonal epidemiological zone, one around the Lake Turkana region, one around the humid and sub-humid belt in Meru County and the other in the semi-arid belt of Kajiado County.

Fig 6 presents a map of malaria exceedance and probabilities, showing areas where p(x)≥ 10% with certainty on the colour gradient.

## Discussion

Understanding the spatial distribution of malaria and the factors that drive its transmission are key in malaria control. Given the heterogeneity of malaria transmission in Kenya, defining the malaria burden at more localized locations is important to allow for targeted control

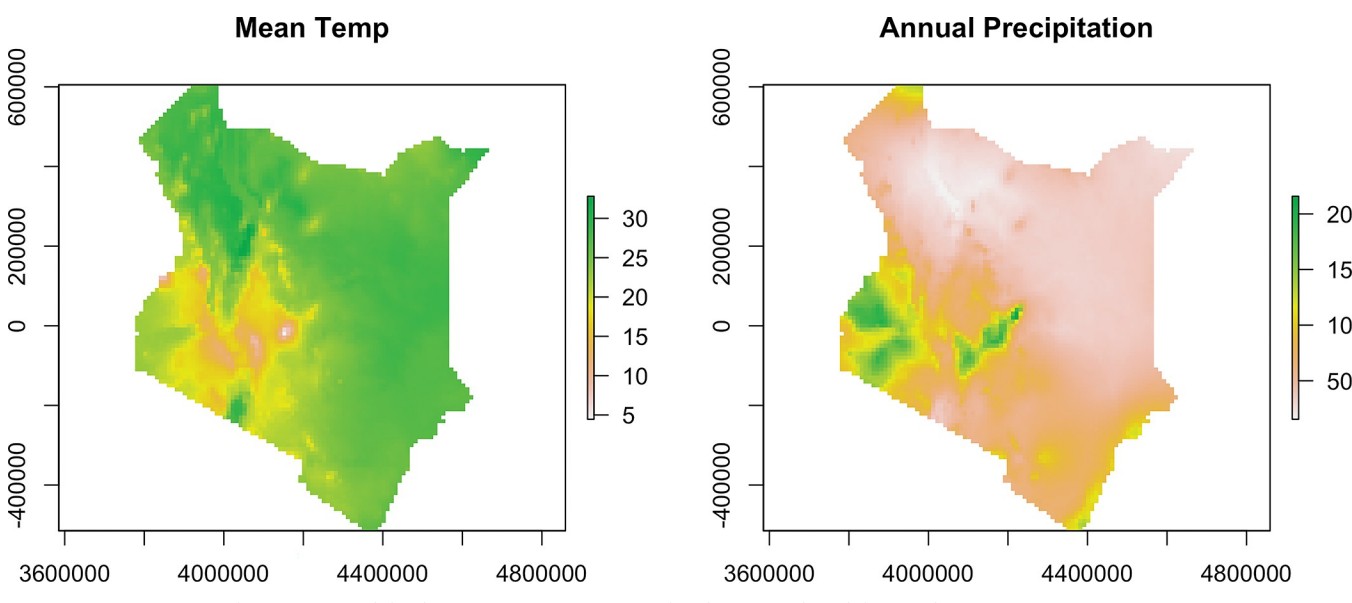

**Fig 3. Mean temperature (degrees Celsius) (top) and Annual precipitation (mm)in Kenya (2020) (Bottom).**

activities. The national malaria indicator surveys performed in the country are not designed to provide malaria prevalence estimates at localized levels. This paper uses Model-Based geostatistical methods to understand malaria transmission drivers in Kenya and map out malaria prevalence at a very high resolution (5 x 5 Km square grid).

In our analysis, we have found that several factors influence malaria transmission. These include gender, age, temperature, rainfall, bed net coverage, elevation, and urbanization. This is consistent with the well-known predictors of malaria transmission. Studies in the same area have previously found a higher risk of malaria among males and an increasing risk with age compared to the first year of life [12]. Both the natural environment and the artificial

**Table 2. Binomial geostatistical model.**

| | Estimate | Standard error | Lower Bound | Upper Bound | p-value |
|---|---|---|---|---|---|
| Intercept | 0.00 | 3.54 | 0.00 | 0.00 | <0.0001 |
| elevation | 1.86 | 1.05 | 1.70 | 2.02 | <0.0001 |
| INT coverage | 3.15 | 1.07 | 2.75 | 3.61 | <0.0001 |
| Urban vs Rural | 0.32 | 1.11 | 0.26 | 0.39 | <0.0001 |
| Rainfall(mm) | 1.91 | 1.06 | 1.69 | 2.15 | <0.0001 |
| Temperature | 1.37 | 1.04 | 1.28 | 1.47 | <0.0001 |
| Age in months (ref <12) | | | | | |
| 12–23 | 1.93 | 1.54 | 0.83 | 4.48 | <0.001 |
| 24–35 | 2.72 | 1.52 | 1.19 | 6.21 | <0.01 |
| 36–47 | 3.56 | 1.51 | 1.59 | 7.96 | <0.001 |
| 48–59 | 7.84 | 1.45 | 3.76 | 16.35 | <0.001 |
| Female vs male | 1.20 | 1.08 | 1.03 | 1.40 | <0.001 |
| Sigma^2* | 0.56 | 1.44 | 0.27 | 1.14 | NA |
| Phi** | 59.57 | 1.85 | 17.87 | 198.50 | NA |
| Tau^2*** | 0.95 | 2.24 | 0.20 | 4.58 | NA |

sigma2 is the variance of the Gaussian process, phi is the scale parameter of the spatial correlation and tau2 is the variance of the nugget effect.

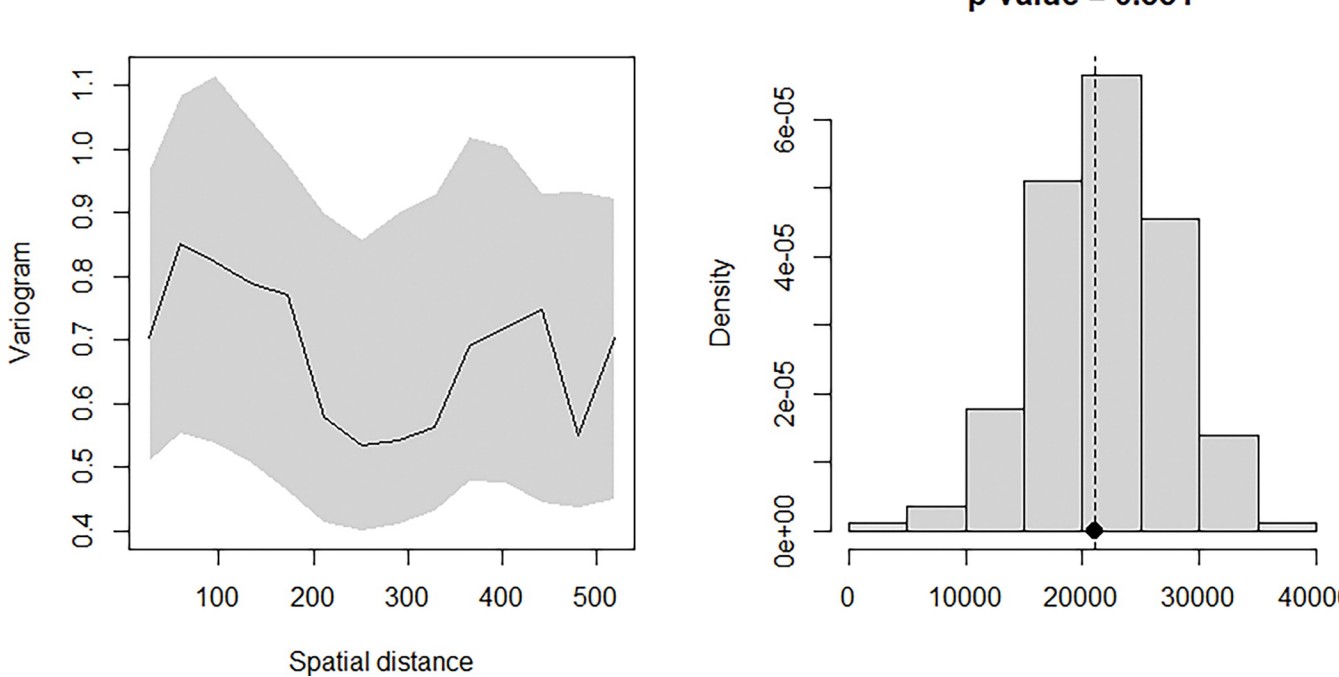

**Fig 4. Model validation.**

environment are known to affect malaria transmission. Temperature, humidity, and rainfall all have interactions with mosquitos at specific points in their life cycle. Temperature regulates the development of mosquitos at each stage. The laying of eggs by mosquitos is reduced in temperature extremes, either too cold or too hot [14]. Temperature also affects the mosquito stage transition, with the optimal temperature being between 22 and 26 degrees Celsius[12].

Rainfall has been shown to be positively correlated with high malaria transmission. During the rainy season, there is usually water logging in the ground, creating mosquito breeding grounds. This analysis observed that prevalence nearly doubles for every mm increase in the annual rainfall. The areas observed to have a higher prevalence of malaria in Kenya are known to have prolonged rainy seasons[6].

The analysis also identified that malaria transmission is higher in rural areas compared to urban areas. This finding is consistent with other studies in the same region. Urban areas may have better housing and improved health services that are easier to access. These factors contribute to the lower risk of malaria. Conversely, rural areas are primarily associated with favourable conditions for malaria, including stagnant water, poor housing, inaccessible health services and agricultural activities [15].

The finding of increasing malaria prevalence with higher bed net coverage can be explained through reverse causality, which is often observed due to the higher distribution of bed nets in areas heavily affected by malaria.

Malaria hotspot areas identified in the analysis include the entire lake and coastal regions classified as malaria endemic [6]. This finding is in keeping with other previous analyses done for past time points [16]. The climatic condition in these areas is known to support malaria transmission. We do find additional hotspots, which highlights the strength in our analysis approach. Localized malaria hotspots are identified in the county of Turkana. Though this area is classified as a seasonal malaria transmission zone, a reactive case detection in the area

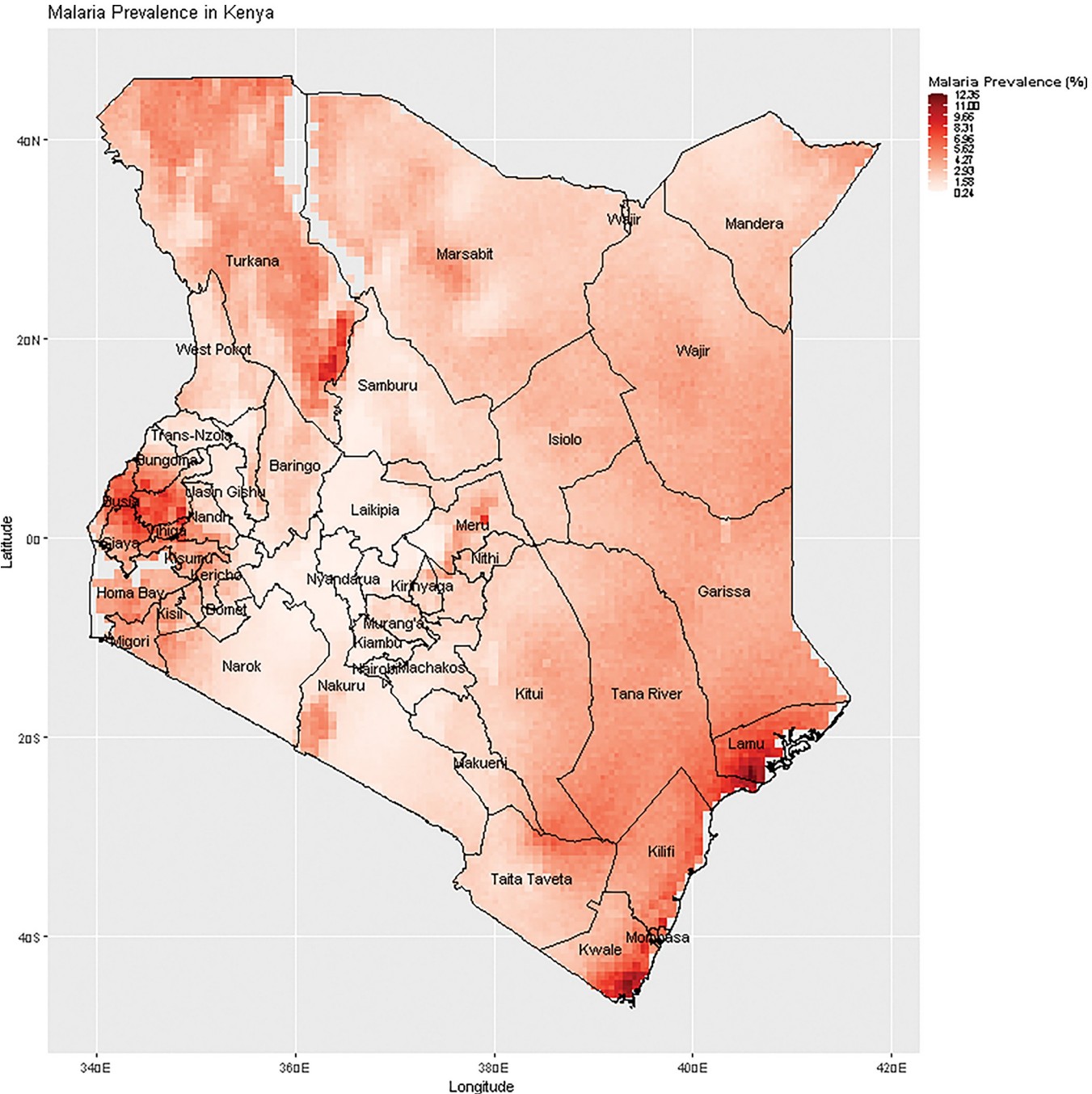

**Fig 5. Malaria prevalence predictions among children six months- 14 years in Kenya.**

conducted from 2018 to 2019 also detected high malaria transmission with a prevalence as high as 33.6% [17]. Another study in a refugee camp in the same region identified a malaria prevalence of 64.2% [17]. Evidence from a recent study examining the contribution of P. falciparum parasite importation to local malaria transmission in Central Turkana confirms that malaria in the area is rather endemic, with intense local transmission as opposed to the importation of malaria [18]. Due to its malaria risk classification status, Turkana is often left out of

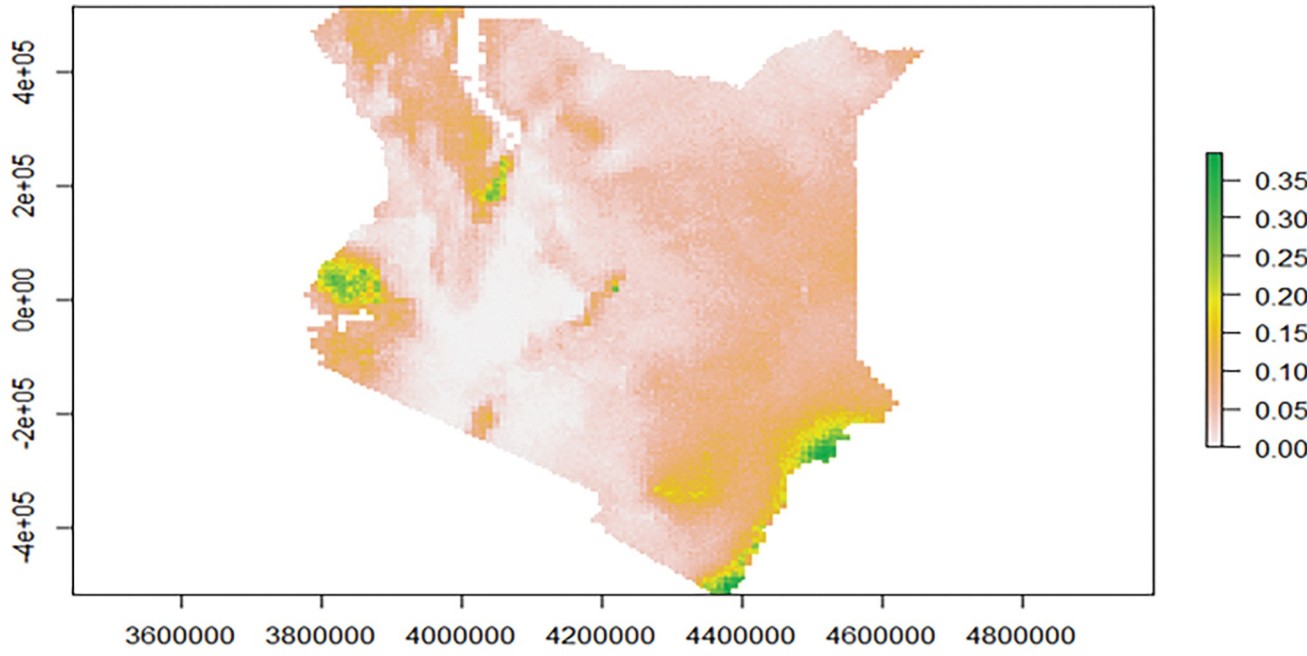

**Fig 6. Map of Kenya showing the exceedance probabilities of malaria prevalence exceeding 10%.**

malaria control activities. This is an important finding where an area's transmission is misclassified. The recent WHO malaria surveillance guide calls for countries to view malaria transmission as a continuum in space and stratify the malaria burden for better targeting and improved efficiency of malaria interventions[8]. As malaria transmission declines, it becomes increasingly focal and prone to outbreaks. Understanding and predicting patterns of transmission risk becomes an essential component of an effective elimination campaign, allowing limited resources for control and elimination to be targeted cost-effectively. By concentrating surveillance, monitoring and control efforts in hotspot areas, programs can maximize the impact of their interventions and reduce the burden of malaria more efficiently. In this study, we also find additional hotspots in the counties of Meru and Kajiado, areas with humid and arid weather conditions, respectively. There is a need for more local surveillance in the area. These areas are also characterized by low implementation of malaria control measures. Future research should prioritize understanding the influence of underlying factors and evaluating the effectiveness of various malaria control interventions in these malaria hotspots.

There are several strengths and limitations of the data used in the analysis. To the best of our knowledge, this is the latest nationally representative data on malaria prevalence. With this, the results of this study are generalizable to the entire population of Kenya. Use of the geostatistical Model as opposed to the traditional non-spatial Model, is a key strength. It allows us to borrow information from the sampled cluster to infer for the unsampled ones and at the same time, account for predictors that influence malaria transmission. The major limitation in the analysis is the lack of adequate environmental covariates to improve spatial predictions.

## Conclusion

The objectives of this analysis were to understand the relationship between malaria prevalence and various predictor factors and examine the disease heterogeneity, including identifying hotspots. Our findings show that rainfall, urbanization, temperature, and bed net coverage are

important factors that affect malaria transmission. The high-resolution malaria prevalence maps produced as part of the analysis are important in identifying hotspots, an essential element in planning, implementation, resource mobilization, monitoring, and evaluation of malaria interventions in the country. We have also identified malaria hotspots in areas not traditionally classified as endemic, highlighting the need to rethink the classification of malaria transmission epidemiology in Kenya.

## Supporting information

**S1 File. The model description for the generalized linear model and the generalized linear mixed model.**
(DOCX)

## Acknowledgments

Many thanks to the DHS programme team for allowing the authors to use the dataset and to the participants that provided the data.

## Author Contributions

**Conceptualization:** Donnie Mategula, Judy Gichuki.

**Data curation:** Donnie Mategula, Judy Gichuki.

**Formal analysis:** Donnie Mategula, Judy Gichuki.

**Funding acquisition:** Donnie Mategula, Judy Gichuki.

**Investigation:** Donnie Mategula, Judy Gichuki.

**Methodology:** Donnie Mategula, Judy Gichuki.

**Project administration:** Donnie Mategula, Judy Gichuki.

**Resources:** Donnie Mategula, Judy Gichuki.

**Software:** Donnie Mategula, Judy Gichuki.

**Supervision:** Donnie Mategula, Judy Gichuki.

**Validation:** Donnie Mategula, Judy Gichuki.

**Visualization:** Donnie Mategula, Judy Gichuki.

**Writing – original draft:** Donnie Mategula, Judy Gichuki.

**Writing – review & editing:** Donnie Mategula, Judy Gichuki.

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
