## [Decision Letter · Decision Letter 0]

21 Aug 2023

PGPH-D-23-01337

Understanding the fine-scale heterogeneity and spatial drivers of malaria transmission using model-based geostatistical methods in Kenya

Dear Mategula,

Thank you for submitting your manuscript to PLOS Global Public Health. After careful consideration, we feel that it has merit but does not fully meet PLOS Global Public Health’s publication criteria as it currently stands. Therefore, we invite you to submit a revised version of the manuscript that addresses the points raised during the review process.

We look forward to receiving your revised manuscript.

Kind regards,

Collins Otieno Asweto, PhD

Academic Editor

Journal Requirements:

2. Some material included in your submission may be copyrighted. According to PLOS’s copyright policy, authors who use figures or other material (e.g., graphics, clipart, maps) from another author or copyright holder must demonstrate or obtain permission to publish this material under the Creative Commons Attribution 4.0 International (CC BY 4.0) License used by PLOS journals. Please closely review the details of PLOS’s copyright requirements here: PLOS Licenses and Copyright. If you need to request permissions from a copyright holder, you may use PLOS's Copyright Content Permission form.

Potential Copyright Issues:

Figure 1-6: please (a) provide a direct link to the base layer of the map (i.e., the country or region border shape) and ensure this is also included in the figure legend; and (b) provide a link to the terms of use / license information for the base layer image or shapefile. We cannot publish proprietary or copyrighted maps (e.g. Google Maps, Mapquest) and the terms of use for your map base layer must be compatible with our CC-BY 4.0 license. 

Reviewers' comments:

Reviewer's Responses to Questions

**Comments to the Author**

1. Does this manuscript meet PLOS Global Public Health’s publication criteria? Is the manuscript technically sound, and do the data support the conclusions? The manuscript must describe methodologically and ethically rigorous research with conclusions that are appropriately drawn based on the data presented.

Reviewer #1: Yes

Reviewer #2: Yes

Reviewer #3: Yes

2. Has the statistical analysis been performed appropriately and rigorously?

Reviewer #1: Yes

Reviewer #2: Yes

Reviewer #3: Yes

3. Have the authors made all data underlying the findings in their manuscript fully available (please refer to the Data Availability Statement at the start of the manuscript PDF file)?

Reviewer #1: Yes

Reviewer #2: No

Reviewer #3: Yes

4. Is the manuscript presented in an intelligible fashion and written in standard English?

Reviewer #1: Yes

Reviewer #2: Yes

Reviewer #3: Yes

5. Review Comments to the Author

Reviewer #1: First :In the abstract, 'ITN' is used as an abbreviation without its full form explicitly mentioned."

Second: In Introduction

In this context, "spatial drivers" refer to the factors or variables that influence the distribution of malaria cases across geographic space. These influential variables, which play a crucial role in shaping the spatial patterns of malaria risk, will be explicitly outlined in the concluding paragraph of the introduction. Although those that were mentioned subsequently.

Third: in methods section : Even though the IRB and DHS acronyms are generally known, they nonetheless require providing explanations before the abbreviation though mentioned later in another paragraph. It should be mentioned first then should be written abbreviation.

Fourth: In the result Section:

The sentence "Malaria Prevalence in five epidemiological zones" could be revised. The term "epidemiological zones" might not be the most commonly used or recognized term in this context. The terminology should be written which are more consistent with standard usage in epidemiological research and might be more appropriate when referring to different areas with distinct epidemiological characteristics.

Fifth: Furthermore, following should be included in the result section as the study focused on Understanding the fine-scale heterogeneity and spatial drivers of malaria transmission using model-based geostatistical methods

1.Geographic Distribution of Malaria Cases: A map displaying the geographical distribution of malaria cases at a fine-scale resolution. This could involve using different colors or shading to represent varying levels of malaria prevalence across different geographic areas. And also mapping of Risk Zones - high-risk or low-risk zones for malaria transmission, these could be delineated on the map.

2. As the study identifies spatial drivers of malaria transmission, you could overlay environmental factors such as temperature, humidity, and rainfall on the map to show how these factors correlate with the spatial distribution of malaria cases.

Reviewer #2: The paper appears to be the introduction and methods section of a scientific research paper focused on understanding the fine-scale heterogeneity and spatial drivers of malaria transmission in Kenya. The paper aims to use model-based geostatistical methods to analyze malaria prevalence data obtained from the Kenya Malaria Indicator Survey (MIS) and investigate the relationship between malaria transmission and various environmental factors. The research also involves creating high-resolution maps of malaria prevalence to identify hotspots and areas of higher transmission risk. I present my comments in three sections, 1. General comments, 2. Specific comments, 3. Comments on the model. Kindly check and address these comments properly.

General Comments:

1. Clarity and Organization: The introduction provides a comprehensive overview of the malaria situation in Kenya and the significance of the study. Consider breaking down the introduction into smaller sub-sections for better readability.

2. Research Objectives: Clearly state the research objectives and hypotheses in the introduction section. This will help readers understand the purpose of the study more explicitly.

3. Citations: There are many sentences, facts, and statistics that are not properly cited. Lines 56-85, the researchers used the same reference. Please consider using multiple references. Most of the article paragraphs are referenced to the same citation (#6).

4. Figure and maps: It is not clear why the researchers added all of these maps. I would recommend removing them and use a specific map with clear lables and purpose (check comment #9).

5. Methods Explanation: The methods section is detailed and comprehensive. However, consider providing more context on the statistical techniques used, especially for readers who may not be familiar with geostatistical models.

6. Variable Definitions: Provide clear definitions of each explanatory variable and its relevance to malaria transmission in Kenya. This will help readers understand the rationale behind the variables chosen for the analysis.

7. Data Source and Cleaning: Discuss any potential limitations or biases in the data source and data cleaning process. This will help readers understand the reliability of the dataset used.

8. Ethics and Consent: While you mentioned obtaining permission to use the dataset and ethical clearance, provide a bit more detail on the ethical considerations, participant consent (if applicable), and data anonymization.

9. Graphical Representation: Use clear, labeled figures and maps that are easy to interpret. Consider providing legends and titles for each figure to enhance their clarity.

Specific Comments:

10. Objectives Clarification: State the specific objectives of the study at the beginning of the "Methods" section. For example, "The objectives of this study were to...".

11. Model Assumptions: Discuss the assumptions underlying the binomial geostatistical model used in the analysis. Highlight any limitations of the model and potential impact on the results.

12. Results Interpretation: In the results section, elaborate on the practical implications of the odds ratios and parameter estimates. Explain how changes in the explanatory variables translate to changes in malaria transmission risk.

13. Model Validation: Provide a brief explanation of the empirical variogram technique used for model validation. This will help readers understand the statistical approach used to assess spatial correlation.

14. Discussion: In the discussion section, relate your findings to existing literature. Discuss similarities and differences with other studies, particularly those conducted in similar settings.

15. Policy Implications: Elaborate on the policy implications of your findings. How can the identified hotspots guide targeted interventions and resource allocation in malaria control programs?

16. Study Limitations: Acknowledge any limitations of your study, such as the availability of certain covariates or potential biases in the data. Discuss how these limitations might affect the interpretation of your results.

17. Future Directions: Suggest potential avenues for future research based on your findings. This could include exploring the effectiveness of specific interventions in the identified hotspots.

18. Conclusion: Summarize the key findings of the study in the conclusion section, relating them back to the research objectives.

Comments on the model:

19. Consider paraphrasing the title of this paragraph: "Model description for the binominal geostatistical Model". You used the word "Model" twice in the title.

20. Clarify Covariate Inclusion: In the model description, provide a clear rationale for the selection of covariates used in the binomial geostatistical model. Explain why elevation, ITN usage, mean temperature, rainfall, and urbanization were chosen as predictor variables. Additionally, elaborate on any potential interactions or collinearity among these covariates.

21. Matérn Correlation Function Parameters: Provide more detailed explanations for the parameters of the Matérn correlation function used in the spatial component of the model. Explain the significance of the scale parameter () and the smoothness parameter () in the context of malaria transmission. This will help readers understand how these parameters influence spatial correlation and variability.

22. Discussion on Variance Parameters: Elaborate on the interpretation and implications of the variance parameters presented in the model. Explain the significance of ² (variance of the Gaussian process), ² (variance of the nugget effect), and their practical implications for the spatial variation of malaria prevalence.

23. Spatial Random Effects: Clarify the concept of spatial random effects (()) and how they capture spatial variation between the sampled clusters. Provide insight into the factors that contribute to the spatial variability, such as geographic features or environmental conditions.

24. Interpretation of Coefficients: In the model formula, explicitly interpret the coefficients associated with each covariate. For example, provide a clear explanation of the interpretation of the coefficient for urbanization (Urban vs Rural) in terms of its effect on the odds of malaria transmission.

25. Inclusion of Interaction Terms: Discuss whether any interaction terms were considered in the model and if they had a significant impact on the results. Interaction terms could capture complex relationships between covariates and provide a more nuanced understanding of their influence on malaria prevalence.

26. Sensitivity to Parameter Choices: Address the sensitivity of the model results to the choices of parameters such as the Matérn correlation function parameters and variance parameters. Explain whether different parameter choices were tested and their potential impact on the findings.

27. Comparison with Alternative Models: Discuss any alternative modeling approaches that were considered and the reasons for choosing the binomial geostatistical model over other options. Compare the strengths and weaknesses of different modeling strategies, if relevant.

28. Implications of Non-Spatial Variation: Explain the potential implications of the Gaussian random variables () representing small-scale spatial variation, measurement error, or within-cluster variation. Discuss how these sources of variation could affect the overall model results and their interpretation.

29. Model Robustness and Validity: Describe how the robustness of the model was assessed. Discuss any sensitivity analyses conducted to examine the impact of different assumptions or parameter choices on the results. Additionally, elaborate on the methods used to validate the model's assumptions and spatial structure.

Reviewer #3: • Congratulations to the researchers. This is a well written manuscript with good statistical analysis to identify disease heterogeneity across Kenya and its policy implications.

• Line 234 indicated that there we 98 clusters in lake endemic areas however in the line 237 (table 1) it was 97. Please check and align.

• Line 239: change ‘’figure one’’ to ‘’Figure 1’’ for consistency.

• To what extent is the misalignment between spatial and especially paucity of environmental covariates affected the validity of the model's results?

6. PLOS authors have the option to publish the peer review history of their article (what does this mean?). If published, this will include your full peer review and any attached files.

**Do you want your identity to be public for this peer review?** For information about this choice, including consent withdrawal, please see our Privacy Policy.

Reviewer #1: No

Reviewer #2: **Yes: **Orwa Al-Abdulla

Reviewer #3: No

---

## [Decision Letter · Decision Letter 1]

1 Nov 2023

Understanding the fine-scale heterogeneity and spatial drivers of malaria transmission using model-based geostatistical methods in Kenya

PGPH-D-23-01337R1

Dear Mategula,

We are pleased to inform you that your manuscript 'Understanding the fine-scale heterogeneity and spatial drivers of malaria transmission using model-based geostatistical methods in Kenya' has been provisionally accepted for publication in PLOS Global Public Health.

Best regards,

Collins Otieno Asweto, PhD

Academic Editor

Reviewer's Responses to Questions

**Comments to the Author**

1. If the authors have adequately addressed your comments raised in a previous round of review and you feel that this manuscript is now acceptable for publication, you may indicate that here to bypass the “Comments to the Author” section, enter your conflict of interest statement in the “Confidential to Editor” section, and submit your "Accept" recommendation.

Reviewer #3: All comments have been addressed

Reviewer #4: All comments have been addressed

2. Does this manuscript meet PLOS Global Public Health’s publication criteria? Is the manuscript technically sound, and do the data support the conclusions? The manuscript must describe methodologically and ethically rigorous research with conclusions that are appropriately drawn based on the data presented.

Reviewer #3: Yes

Reviewer #4: Yes

3. Has the statistical analysis been performed appropriately and rigorously?

Reviewer #3: Yes

Reviewer #4: Yes

4. Have the authors made all data underlying the findings in their manuscript fully available (please refer to the Data Availability Statement at the start of the manuscript PDF file)?

Reviewer #3: Yes

Reviewer #4: Yes

5. Is the manuscript presented in an intelligible fashion and written in standard English?

Reviewer #3: Yes

Reviewer #4: Yes

6. Review Comments to the Author

Reviewer #3: Congratulations to the research team.

Reviewer #4: The authors did great job responding to the comments by the reviewers. As the fourth reviewer, I think most of my concerns have been addressed. However, line 65 & 66 highlighted the decline in Malaria prevalences in previous years 13%, 8% for 2010 and 2015 respectively but the prevalence for year 2020 is missing.

7. PLOS authors have the option to publish the peer review history of their article (what does this mean?). If published, this will include your full peer review and any attached files.

**Do you want your identity to be public for this peer review?** For information about this choice, including consent withdrawal, please see our Privacy Policy.

Reviewer #3: **Yes: **Richard AMENYAH

Reviewer #4: No
